# Efficacy of Combined Initial Treatment of Methotrexate with Infliximab in Pediatric Crohn’s Disease: A Pilot Study

**DOI:** 10.3390/biomedicines11092575

**Published:** 2023-09-19

**Authors:** Yoon-Zi Kim, Ben Kang, Eun-Sil Kim, Yiyoung Kwon, Yon-Ho Choe, Mi-Jin Kim

**Affiliations:** 1Department of Pediatrics, Samsung Medical Center, Sungkyunkwan University School of Medicine, Seoul 06351, Republic of Korea; 2Department of Pediatrics, School of Medicine, Kyungpook National University Chilgok Hospital, Daegu 41944, Republic of Korea; 3Department of Pediatrics, Kangbuk Samsung Medical Center, Sungkyunkwan University School of Medicine, Seoul 03181, Republic of Korea; 4Department of Pediatrics, Inha University Hospital, Inha University School of Medicine, Incheon 22188, Republic of Korea; na202@inha.ac.kr

**Keywords:** pediatric Crohn’s disease, immunogenicity, azathioprine, methotrexate, trough level of anti-TNF agent, anti-drug antibody, immunomodulatory

## Abstract

Background: The combination of antitumor necrosis factor-alpha (TNF-α) agents with immunomodulators (IMMs) is a common treatment for pediatric Crohn’s disease (CD). Although methotrexate (MTX) can be a first-line medication as an IMM, most clinicians in real-life practice, especially in South Korea, are more familiar with thiopurines. This study aimed to compare the efficacy and immunogenicity of MTX and azathioprine (AZA) as concurrent therapies for pediatric CD. Methods: In this pilot study, 29 newly diagnosed pediatric patients with moderate-to-severe CD were randomized to receive either MTX (*n* = 15) (15 mg/body surface area (BSA) per week) or oral AZA (*n* = 14) (0.5 mg/kg per day) in combination with Infliximab (IFX). The primary outcomes were the proportion of patients in endoscopic, biochemical, and transmural remission after 14 and 54 weeks of IFX therapy. The trough levels (TLs) of IFX and anti-drug antibody (ADA) levels were also compared. Results: Among the 29 patients, there were no significant differences in the biochemical (*p* = 1.0 at week 14, *p* = 0.45 at week 54), endoscopic (*p* = 0.968 at week 14, *p* = 0.05 at week 54), or transmural (*p* = 0.103 at week 54) remission rates between the two medications during the concurrent therapy. Additionally, the trends in the IFX trough and ADA levels over time during the treatments were similar for both medications, with no significant differences (*p* = 0.686, *p* = 0.389, respectively). Conclusion: The MTX showed comparable efficacy to the AZA in pediatric CD patients with moderate-to-severe disease. This effectively maintained adequate IFX levels and reduced ADA production. Therefore, although additional large-scale clinical trials are needed, this study demonstrated that either MTX or AZA can be selected as IMMs in the concurrent treatment of pediatric CD, depending on individual medical institutions’ circumstances.

## 1. Introduction

In Crohn’s disease (CD), the combination of anti-tumor necrosis factor-alpha (TNF-α) agents and immunomodulators (IMMs) such as methotrexate (MTX) and azathioprine (AZA) is known to increase instances of complete remission and to prevent recurrence [1]. However, a significant proportion of CD patients (ranging from 37.8% to 43.3%) undergoing anti-TNF-α treatment experience a loss of response (LOR). As a result, they require interventions such as dose intensification, switches to different anti-TNF agents, or a switch to a medication with a different mechanism of action [2]. Notably, LOR can be partially explained by pharmacokinetic problems and subtherapeutic drug concentrations caused by the development of anti-drug antibodies (ADAs), a phenomenon known as immunogenicity [3]. 

The development of ADAs can reduce the efficacy of anti-TNF-α agents by either neutralizing the biologic or facilitating the clearance of the drug [4,5]. Multiple studies suggest that the combination of anti-TNF-α agents with IMMs may reduce ADA development, thereby maintaining a state of remission [6,7,8,9,10,11]. Notably, the most frequently used IMMs for inflammatory bowel disease (IBD) are thiopurines, such as AZA or 6-mercaptopurine [12,13]. As with other IMMs, MTX can be used alone or in combination therapy for CD rather than for ulcerative colitis (UC) [14,15]. However, most clinicians, especially in South Korea, treating pediatric patients with CD are more familiar with thiopurines, and the use of MTX is somewhat less common in many settings because reports on combination treatment with MTX are relatively scarce (AZA use in CD was 92.7% vs. MTX use in CD was 1.8%). [16] Therefore, this study aimed to evaluate the efficacy of an initial combination treatment of MTX and IFX in comparison to AZA in pediatric CD patients. 

## 2. Materials and Methods

### 2.1. Patients

This randomized prospective pilot study was conducted at the Department of Pediatrics, Samsung Medical Center, between January 2020 and January 2022. Eligible participants for this study were individuals between the ages of 6 and 20 years who had recently been diagnosed with moderate-to-severe pediatric CD, had a Pediatric Crohn’s Disease Activity Index (PCDAI) ≥30, and had not been treated with anti-TNF-α agents or IMMs. The diagnosis of CD was confirmed by clinical, endoscopic, and pathological criteria according to the revised Porto criteria of the European Society for Pediatric Gastroenterology, Hepatology, and Nutrition [17], and the disease-phenotype classification was based on the Paris classification [18].

Patients diagnosed with UC or unspecified inflammatory bowel disease were excluded. Additionally, patients with a history of cancer, tuberculosis, or intestinal stenosis requiring emergency or elective surgery were also excluded.

### 2.2. Treatment Regimens

Patients who agreed to participate in the trial were randomized to receive either MTX or AZA as an IMM. The MTX (MTX group) was administered at a dose of 15 mg/BSA per week. The drug was administered subcutaneously at the first dose and was switched to an oral tablet after the second dose at the same dosage. The AZA (AZA group) was administered as an oral tablet at a dose of approximately 0.5 mg/kg per day. The dose was increased if the patient did not have a genetic mutation in thiopurine metabolism (such as in the thiopurine methyltransferase (*TPMT*) or nudix hydrolase 15 (*NUDT15*) genes). During the study period, the AZA dose was carefully adjusted according to the thiopurine metabolite (6-thioguanine nucleotide (6-TGN) and 6-methylmercaptopurine (6-MMP)) monitoring system. In the Korea Health Insurance Review and Assessment Service, pediatric CD patients with PCDAI ≥30 must have a history of using at least two drugs out of steroids, 5-ASA, or IMM for at least 3 months in order for biologics to be covered by insurance. Because of this insurance practice, the patients in this study received 5-ASA and an IMM when the diagnosis was confirmed. As 5-ASA, mesalazine was administered to all patients at doses of 40–80 mg/kg/day concurrently with AZA and MTX. 

All patients received induction therapy with IFX (SB2, Remaloce^®^ (Samsung Bioepies, Incheon, Republic of Korea)) as the anti-TNF-α agent (infused intravenously 5 mg/kg/dose). The IFX was administered as an induction regimen of 5 mg/kg at 0, 2, and 6 weeks, and scheduled maintenance therapy with IFX was repeated every 8 weeks without dose intensification. Dropout from the study was considered at the discretion of the researcher if IFX-dose intensification was required by LOR during the study. The use of steroids was not permitted in this study, as they may affect remission. However, subjects who were on steroid reduction at the time of enrolment were allowed to enroll in the study. 

### 2.3. Data Collection, Study Design, and Follow-Up

Patients were examined at baseline, after induction therapy (week 14), and 1 year (week 54) after treatment with IFX. At diagnosis, baseline demographic and clinical data, including sex, age, and disease phenotype, were collected from electronic medical records. At each visit, each patient’s weight, height, blood pressure, and body temperature were measured, and disease activity was evaluated based on the PCDAI. In addition, blood samples were collected for complete blood count (CBC), erythrocyte-sedimentation rate (ESR), C-reactive protein (CRP), serum-albumin, serum-amylase, lipase, and liver-function tests. Fecal calprotectin levels were also evaluated. Ileocolonoscopy was performed at diagnosis, after induction therapy (week 14), and one year (week 54) after treatment with IFX. As an additional analytic method, magnetic resonance enterography (MRE) was performed twice: at diagnosis and after 1 year of treatment. 

Clinical data were collected at week 14 and week 54 after the initiation of IFX, including IFX trough levels (TLs) and ADA levels. The TL of IFX was assessed using an IDK monitor^®^ IFX drug-level enzyme-linked immunosorbent assay (ELISA) kit (Immundiagnostik AG, Bensheim, Germany). For ADA detection, the IDK monitor^®^ IFX free ADA ELISA (Immundiagnostik AG, Bensheim, Germany) kit was used. The results were interpreted using a cut-off control (10 AU/mL). 

Reasons for patient withdrawal included non-compliance, severe infection, repeated abnormal liver-function-test results, persistent leukopenia (<3000/mm^3^), thrombocytopenia (<100,000/mm^3^), marked worsening of symptoms, unacceptable side effects, or patient requests. Patients who withdrew from the medications due to adverse events were followed up in the same manner as those who continued with the medications. 

### 2.4. Outcome, Measures, and Definitions

The primary outcomes were the clinical, biochemical, endoscopic, and transmural remission rates after induction therapy and 1 year of treatment. Clinical remission was defined by a PCDAI score < 10 points [19], and biochemical remission was defined by CRP levels <0.3 mg/dL and fecal calprotectin levels ≤150 ug/g at each scheduled visit. Transmural remission was defined as a thickness < 3 mm in the absence of ulcers, edema, enhancement, and complications for all ileocolonic segments, as evaluated using MRE. Endoscopic remission was defined as a Simple Endoscopic Score for CD (SES-CD) <3, which was calculated as the sum of the selected endoscopic parameters (ulcer size, ulcerated and affected surfaces, and stenosis). 

The following secondary outcome measures were compared: TLs of IFX and ADA levels between the two groups after 1 year of treatment. 

### 2.5. Statistical Analysis

Student’s *t*-test was used for continuous variables, while Fisher’s exact test or the χ^2^ test was used for discrete variables. The mean ESR, CRP, fecal calprotectin values, and TLs of IFX and ADA levels were compared using repeated measures analysis of variance. The proportions of patients with clinical, biochemical, transmural, and endoscopic remission were compared using Fisher’s exact test. Univariate Cox proportional hazard regression analyses were used to investigate factors associated with biochemical, endoscopic, and transmural remission. Factors with *p* < 0.1 in the univariate analyses were included in the multivariate analyses. The results are expressed as hazard ratio (HR) with 95% confidence interval (CIs). Kaplan–Meier analysis was used to calculate biochemical remission, and the log-rank test was used to determine the overall statistical difference in estimates. 

All statistical tests were conducted at a significance level of 0.05. All statistical analyses were performed using SPSS version 27 (IBM Corp., Armonk, NY, USA). 

## 3. Results

### 3.1. Baseline Characteristics

From January 2020 to January 2022, 34 patients were screened for the study. Of these, four patients who did not meet the inclusion criteria were excluded, and one patient was withdrawn from the study owing to insurance issues that prevented the continued infusion of IFX. In total, 29 patients were randomly assigned to receive either MTX or AZA therapy. 

The median age was 13.48 years in the MTX group and 13.16 years in the AZA group, and the majority of the patients were male. The baseline characteristics of the two groups (Table 1) did not exhibit any significant differences. At baseline, the initial average doses of MTX and AZA were 14.66 mg/BSA (14.13–15.2 mg/BSA) and 0.53 mg/kg (0.49–0.56 mg/kg), respectively. 

After 1 year of treatment, the average dose of MTX was 12.26 mg/BSA (10.01–14.43 mg/BSA), while that for AZA was 0.66 mg/kg (0.57–0.76 mg/kg). In addition, the mean 6-TGN level after 1 year of treatment was 245.2 ± 68.85 pmol/8 × 10^8^ RBC (an appropriate TL: 230–450 pmol/8 × 10^8^ RBC). By conducting *TPMT*/*NUDT15* gene testing, it was determined that three patients in the MTX group (one *TPMT* mutation and two *NUDT15* mutations) and five patients in the AZA group (one *TPMT* and four *NUDT15* mutations) were heterozygous.

### 3.2. Outcomes

#### 3.2.1. Primary Outcome Measures 

After induction, there were no statistically significant differences in the proportions of patients in biochemical remission (*p* = 1.0; 95% CI, 0.238–4.198) or endoscopic remission (*p* = 0.968; 95% CI, 0.16–5.795). In terms of biochemical (*p* = 0.45, 95% CI, 0.102–2.305), endoscopic (*p* = 0.05, 95% CI, 0.025–0.950), and transmural remission (*p* = 0.103, 95% CI, 0.807–32.773), there were no significant differences between the two groups after 1 year of IFX therapy (Figure 1 and Figure 2).

#### 3.2.2. Secondary-Outcome Measures 

The mean CRP and fecal calprotectin levels during the study period are shown in Figure 3. There were no statistically significant differences between the two groups at baseline (CRP: AZA group 0.284 ± 0.093, MTX group 0.557 ± 0.09, *p* = 0.13; fecal calprotectin: AZA group 1440.73 ± 239.265, MTX group 1616.97 ± 239.265, *p* = 0.529), after induction (CRP: AZA group 0.071 ± 0.09, MTX group 0.145 ± 0.09, *p* = 0.252; fecal calprotectin: AZA group 305.533 ± 239.265, MTX group 480.867 ± 239.265, *p* = 0.425), and after one 1 year of treatment (CRP: AZA group 0.11 ± 0.093, MTX group 0.137 ± 0.099, *p* = 0.506; fecal calprotectin: AZA group 634.821 ± 247.44, MTX group 370.831 ± 266.783, *p* = 0.505).

The trends in the IFX TLs over time during the treatment periods were similar for both medications, without any significant differences (Figure 4) (MTX group 5.68 ± 3.23 ug/mL, AZA group 5.11 ± 3.61 ug/mL, *p* = 0.686 at week 54). Similarly, in terms of the ADA levels, both drugs also showed similar trends, without any significant differences (MTX group 6.45 ± 5.61 AU/mL, AZA group 3.91 ± 2.12 AU/mL, *p* = 0.389 at week 54).

The univariate Cox proportional hazard regression analyses showed that risk factors, including two medications, were associated with endoscopic (HR: 0.71, 95% CI: 0.13–3.90, *p* = 0.691) (Table 2), biochemical (HR: 0.71, 95% CI: 0.11–4.55, *p* = 0.714), and transmural remission (HR: 0.77, 95% CI: 0.45–1.34, *p* = 0.358). 

The Kaplan–Meier survival curves were also calculated, and there were no significant differences between the endoscopic (log-rank test *p* = 0.468, Figure 5), biochemical (log-rank test *p* = 0.557), or transmural remission rates (log-rank test *p* = 0.126) in the MTX and AZA groups.

### 3.3. Safety

In terms of safety, two patients in the MTX group withdrew from treatment as a result of adverse events (one case of acute pancreatitis and one case of leukocytopenia). Additionally, the symptoms of two patients who experienced nausea and vomiting while on MTX improved after dose reductions. Meanwhile, one patient in the AZA group presented with leukocytopenia, which returned to normal after the withdrawal of the medication.

## 4. Discussion

This study aimed to evaluate the efficacy of concurrent treatment with MTX and AZA as first-line IMMs for pediatric CD patients during treatment with IFX. Although the therapeutic effect of the IFX may have masked the efficacy of the two IMMs, there were no significant differences in biochemical, endoscopic, and transmural remission rates between the two IMMs. The treatment was also equally effective in maintaining adequate IFX TLs and reducing the production of ADAs.

The remission rates observed in this study were higher than those reported in previous research. Various factors, such as patient selection, dose, and the route of the drugs’ administration, my account for these differences. Moreover, at our medical center, we mainly applied a top-down strategy of early use of IFX with IMMs in patients with newly diagnosed CD, which is known to be more effective in achieving remission; therefore, this could account for the higher remission rates [20,21,22].

Among the various types of IMM, several studies have investigated the efficacy of MTX in pediatric CD [23]. Evidence has revealed that not only is MTX an effective IMM for maintaining remission in pediatric patients with CD, but also that it offers additional benefits. Specifically, MTX has shown efficacy in treating thiopurine-resistant pediatric CD patients and has demonstrated the ability to promote complete mucosal healing [24,25]. Nevertheless, the number of published studies on the effect of MTX in achieving remission in moderate-to-severe CD is still limited compared to of the number of equivalent studies on AZA. Therefore, most gastroenterologists treating patients with pediatric CD are more familiar with thiopurines, and the use of MTX is less common in many settings. Unfortunately, the prescription of medications such as thiopurine, which carry the potential risk of malignancy, is always a concern and burden for healthcare providers. Kandiel et al., suggested an approximate fourfold increased risk of lymphoma in IBD patients treated with AZA [26]. Additionally, in one meta-analysis, the standardized incidence ratio (SIRs) for lymphoma in IBD exposed to thiopurines was 4.92 (95% CI, 3.10–7.78), and the level of risk became significant after 1 year of exposure. Furthermore, it has been observed that men are at greater risk than women (SIR for men = 4.50; 95% CI: 3.71–5.40; SIR for women = 2.29; 95% CI: 1.69–3.05, relative risk = 1.98; *p* < 0.05). Moreover, patients younger than 30 years had the highest relative risk (SIR = 6.99; 95% CI, 2.99–16.4), which is a major concern for pediatric gastroenterologists, as they may need to prescribe medications with potential malignancy risks to individuals at a young age [27].

Considering the data highlighting the potential risk of malignancy associated with thiopurine use, MTX has re-emerged as a first-line IMM, as there has been no increase in the incidence of malignancy or lymphoma reported in association with MTX to date [28]. Notably, some guidelines for pediatric luminal CD recommend the use of MTX over thiopurines, particularly in male patients [29,30].

In addition to the potential risk of malignancy, another common side effect of AZA is leukocytopenia. A large series of studies reported that AZA led to bone-marrow toxicity in 5% of the population (manifesting within a median of 12.5 months, with the duration ranging from 2 weeks to 11 years during treatment), which could potentially lead to severe outcomes such as sepsis or death [31,32]. In our study, one patient withdrew from the AZA treatment due to leukocytopenia. Fortunately, this patient recovered immediately after their discontinuation of the medication. To alleviate thiopurine-induced leukopenia, it is conventionally recommended to assess each patient’s *TPMT* and *NUDT15* gene status prior to initiating thiopurine therapy. Notably, approximately 23.8% of Koreans have been reported to have *NUDT15* heterozygote or homozygote mutations, which is a relatively high proportion compared to that of the European population (approximately 0.5%). Moreover, even at low initial AZA doses (0.5–1.0 mg/kg/day), early leukocytopenia within 4 weeks of the initiation of AZA treatment was observed in patients with homozygous *NUDT15* mutations [33,34]. Therefore, it is crucial to perform genotyping when initiating thiopurine treatment for IBD. Furthermore, because AZA has a narrow therapeutic window, it is generally recommended to monitor the levels of 6-TGN and 6-MMP, which are the active metabolites of AZA, in IBD patients treated with AZA [35,36]. Unfortunately, genetic assessments and monitoring systems are not available in all medical institutions. In real-life practice, clinicians at medical institutions must often rely on requesting genotyping and metabolite-level tests from central laboratories, which can result in a significant delay of several months before the results are received. This prolonged waiting period poses challenges in the delicate management of patients, making it difficult to promptly adjust treatment plans based on patients’ genetic and metabolite information.

In comparison to AZA, similar bone-marrow suppression has also been observed with MTX, along with acute pancreatitis, nausea, and vomiting. However, the incidence of bone-marrow suppression with MTX appears to be lower than that with AZA [37]. It has been observed that most of the side effects of MTX can be managed by reducing the dose, indicating that the occurrence of these events is dependent of the dose [38]. Moreover, severe bone-marrow suppression is associated with low folate levels in the absence of folate supply, low serum-albumin concentrations, and renal impairment. Notably, low levels of serum albumin and decreased kidney function lead to a decrease in MTX clearance and, eventually, to an increase in MTX levels and MTX metabolites, leading to bone-marrow toxicity [39]. These side effects of MTX can be controlled by supplying folate and monitoring serum-albumin and -creatinine levels, which are readily available in most medical institutions [40].

Currently, many clinicians are focusing on the role of IMMs in reducing immunogenicity, specifically in terms of ADA production for biological therapy [6,41]. Indeed, numerous data support the role of IMMs in reducing the immunogenicity of anti-TNF-α agents (IFX or adalimumab) in patients receiving combination therapy [42]. However, the mechanisms of action of IMMs in suppressing the immunogenicity of anti-TNF-α agents have not yet been fully elucidated. Nonetheless, it has been suggested that MTX and thiopurines eliminate ADAs by suppressing T-cell proliferation and inducing the attenuation of memory B cells and CD-4T cells [43]. Additionally, one hypothesis suggests that MTX induces immune modulation by reducing early B- and T-cell responses to biological agents [44,45]. However, further research is required to better understand the mechanisms of action of IMMs in terms of immunogenicity.

To date, no head-to-head comparative studies have been conducted on the efficacy of MTX and AZA in reducing the risk of immunogenicity to IFX in pediatric patients with CD. Therefore, this study attempted to address the issue of whether MTX reduces the risk of immunogenicity by measuring IFX TLs and ADA levels during treatment in comparison to AZA. Although it is unclear which drug reached the ideal therapeutic level first, the trends in IFX TL over time during treatment were similar for both medications. Given that ADA levels rise over time in pediatric CD patients undergoing IFX therapy [46], both drugs showed a similar tendency to increase ADA levels, without any significant differences. This suggests that MTX has similar immunogenicity to AZA. Furthermore, although it was difficult to statistically compare the slopes in Figure 4b because the repeated-measures analyses of variance were performed at two time points, the ADA levels increased more rapidly over time in the MTX group. However, it is important to interpret these findings with caution, as the ADA levels measured in this study represent the concentration of total antibodies and may not directly reflect the presence of neutralizing antibodies, which are considered to be more reliable predictors of LOR.

A significant limitation of this study is the relatively small sample size of pediatric patients, which hindered our ability to draw definitive conclusions. Although this was a pilot study, there were obstacles to enrolling sufficient pediatric patients in this prospective study due to parents’ concerns. Considering that the recommended doses of AZA for the treatment of pediatric CD are 2.0–2.5 mg/kg/day in patients with normal *TPMT* metabolism, the dose of AZA administered in this study was relatively low [47]. In terms of the AZA dose, a dose-escalating strategy tends to be used in Asian countries to stratify AZA therapy, starting with lower doses of 0.5–1.0 mg/kg/day, and subsequently carefully adjusting the dose to the target dose according to the thiopurine-metabolite level, as well as genetic testing results [48,49,50]. The main reason for the different AZA-dosing regimens in Asian countries is the high incidence of thiopurine-induced leukocytopenia in Asian patients, even at lower doses than those used in Western countries [51,52]. Moreover, in terms of maintenance therapy, there is cumulative evidence for therapeutic efficacy even at a low dose of AZA (1.0–1.5 mg/kg/day) in treating IBD in Asian populations [53,54,55]. In fact, even the low dose of AZA in this study maintained clinical remission with an adequate therapeutic level of thiopurine metabolite. Furthermore, considering that IBD is a chronic disease, the relatively short follow-up period of this study may limit our understanding of long-term treatment responses.

## 5. Conclusions

In this study, MTX was shown to have a therapeutic efficacy comparable to that of AZA in pediatric patients with moderate-to-severe CD receiving IFX. It was also equally effective in maintaining adequate IFX TLs and reducing the production of ADAs. Therefore, although additional large-scale clinical trials are needed, this study demonstrated that either MTX or AZA can be selected as IMMs in the concurrent treatment of pediatric CD, depending on individual medical institutions’ circumstances.

## Figures and Tables

**Figure 1 biomedicines-11-02575-f001:**
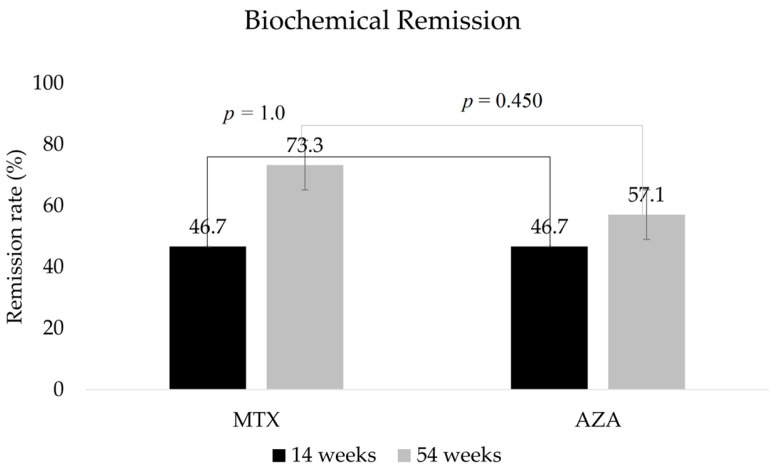
Proportions of patients by study group in biochemical remission after 14 and 54 weeks of therapy. MTX, methotrexate; AZA; azathioprine.

**Figure 2 biomedicines-11-02575-f002:**
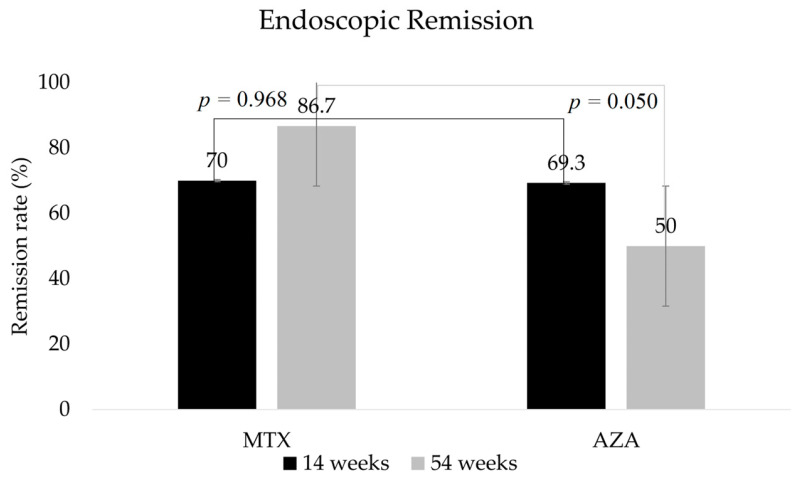
Proportions of patients by study group in endoscopic remission after 14 and 54 weeks of therapy. MTX, methotrexate; AZA; azathioprine.

**Figure 3 biomedicines-11-02575-f003:**
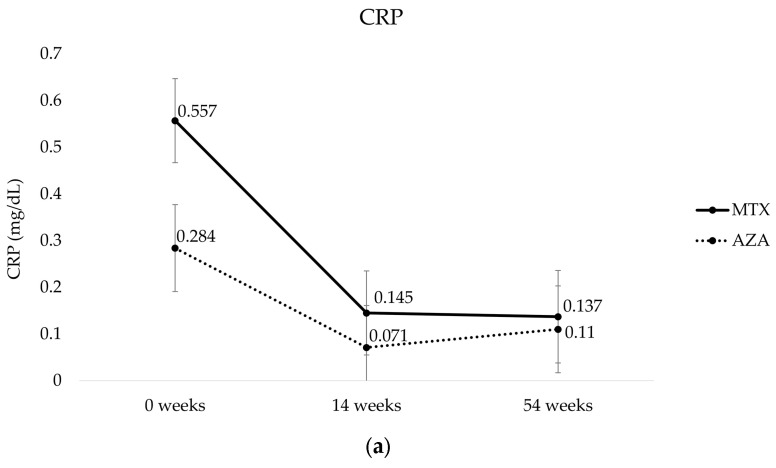
Mean ± SD values for (**a**) CRP and (**b**) fecal calprotectin at baseline, after 14 and 54 weeks of therapy. The *p*-values for the overall comparisons between the two groups were derived from a repeated-measures analysis of variance. MTX, methotrexate; AZA; azathioprine; CRP, C-reactive protein.

**Figure 4 biomedicines-11-02575-f004:**
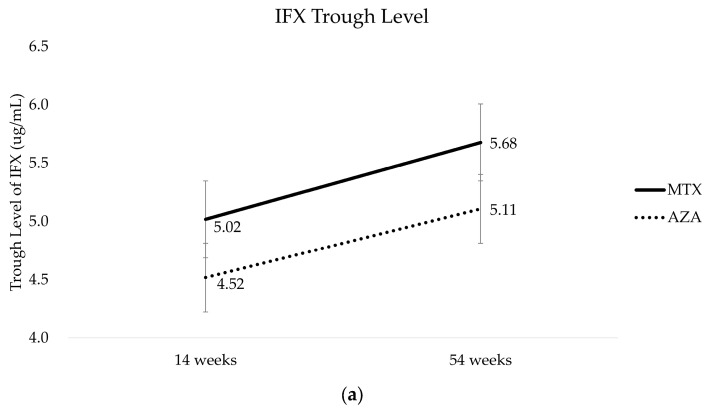
(**a**) Mean IFX TLs by study group after 14 and 54 weeks of therapy. (**b**) Mean ADA levels by study group after 14 and 54 weeks of therapy. MTX, methotrexate; AZA; azathioprine; IFX, Infliximab.

**Figure 5 biomedicines-11-02575-f005:**
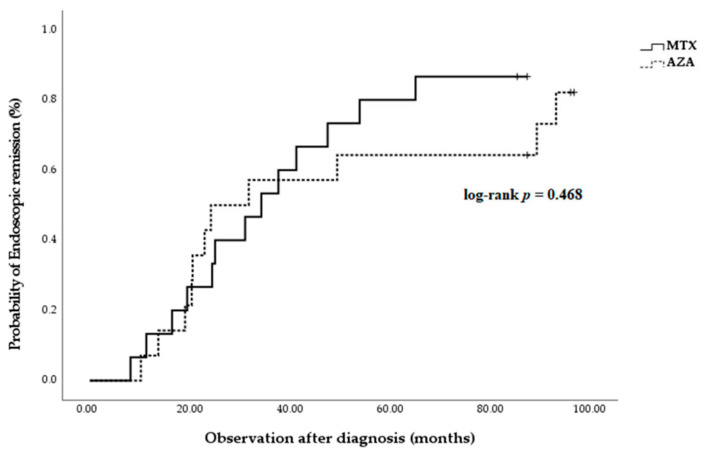
Kaplan–Meier survival curves for endoscopic remission between two groups. There were no significant differences in endoscopic remission between the two medications (log-rank test *p* = 0.468). MTX, methotrexate; AZA; azathioprine.

**Table 1 biomedicines-11-02575-t001:** Baseline patient characteristics upon the initial diagnosis of Crohn’s disease.

		MTX Group	AZA Group	*p*-Value
Patients (*n*)		15	14	
Age (years)		13.48 (11.61–15.36)	13.16 (11.13–15.20)	0.770 ^a^
Male sex (%)		14 (93.3%)	11 (78.6%)	0.331 ^b^
BMI		19.46 (17.38–21.55)	19.09 (17.43–20.75)	0.496 ^a^
Hematocrit (%)		42.05 (4013–43.95)	39.73 (38.02–41.44)	0.065 ^a^
Albumin (g/dL)		4.43 (4.5–4.1)	4.33 (4.20–4.47)	0.237 ^a^
ESR (mm/hr)		9.47 (3.13–15.81)	17.42 (9.52–25.32)	0.095 ^a^
CRP (mg/dL)		0.27 (0.11–0.43)	0.56 (0.19–0.92)	0.130 ^a^
Calprotectin (mg/kg)		1440.73 (723.87–2157.59)	1616.97 (1089.97–2143.97)	0.529 ^a^
PCDAI ^c^		30.38 (28.73–32.03)	31.54 (30.36–32.72)	0.244 ^a^
Initial dose of the drug		14.66 mg/BSA (14.13–15.2)	0.53 mg/kg (0.49–0.56)	
Dosage of the drugs at 54 weeks		12.26 mg/BSA (10.01–14.43)	0.66 mg/kg (0.57–0.76)	
Dose of mesalazine (mg/kg)		46.08 (41.4–50.76)	43.93 (39.57–48.28)	0.210 ^a^
*TPMT* mutations (patient number)		1 (6.7)	1 (7.1)	
*NUDT15* mutations (patient number)		2 (13.3)	4 (28.6)	
Paris classification at diagnosis				
Age at diagnosis	A1a	2 (13.3)	4 (28.6)	
	A1b	11 (73.3)	5 (35.7)	
	A2	2 (13.3)	5 (35.7)	
Location	L1	2 (13.3)	1 (7.1)	
	L2	1 (6.7)	1 (7.1)	
	L3	13 (86.7)	12 (85.7)	
	L4a	6 (40)	3 (21.4)	
	L4b	2 (13.3)	4 (28.6)	
	L4ab	7 (46.7)	7 (50)	
Behavior	B1	15 (100)	14 (100)	
	B2	0	0	
	B3	0	0	
	p	13 (86.7)	10 (71.4)	

Values are presented as median (interquartile range) or number (%). ^a^: independent-sample *t*-test, ^b^: Mann–Whitney *U* test, ^c^: Pediatric Crohn’s Disease Activity Index (PCDAI) score ranging from 0 to 100, with higher scores indicating more active disease. A score <10 indicates inactive disease, 11–30 indicates mild disease, and >30 indicates moderate-to-severe disease. BMI, body-mass index; ESR, erythrocyte-sedimentation rate; CRP, C-reactive protein; PCDAI, Pediatric Crohn’s Disease Activity Index; *TPMT*, thiopurine methyltransferase; *NUDT15*, nudix hydrolase 15. The classification in the table is as follows. A1a: age <10 years, A1b: 10 to <17 years, A2: age ≥ 17 years, L1: distal 1/3 ileum ±limited cecal disease, L2: colonic disease, L3: ileocolonic disease, L4a: upper disease proximal to ligament of Treitz, L4b: upper disease distal to the ligament of Treitz and proximal to the distal 1/3 ileum, L4ab: upper disease involvement in both L4a and Lab. B1: non-stricturing, non-penetrating behavior, B2: stricturing behavior, B3: penetrating behavior, P: perianal disease modifier.

**Table 2 biomedicines-11-02575-t002:** Cox proportional hazard regression analysis of factors associated with endoscopic remission in patients with CD.

	Univariate Cox Analysis		
	HR	95% CI	*p*
Sex (female vs. male)	0.38	0.41–3.42	0.385
Age at diagnosis	1.07	0.84–1.37	0.583
Any colonic involvement	1.84	0.84–2.67	0.990
Any upper gastrointestinal involvement	1.60	0.25–10.29	0.619
PCDAI at diagnosis	1.00	0.98–1.02	0.942
Erythrocyte sedimentation rate at diagnosis	1.05	0.97–1.14	0.193
Albumin at diagnosis	3.49	0.08–120	0.514
C-reactive protein at diagnosis	0.41	0.07–2.45	0.326
SES-CD at diagnosis	1.00	0.84–1.01	0.078
MTX vs. AZA	0.71	0.13–3.90	0.691

HR, hazard ratio; CD, Crohn’s disease; PCDAI, Pediatric Crohn’s Disease Activity Index; SES-CD, simple endoscopic score for Crohn’s disease; MTX, methotrexate; AZA, azathioprine.

## Data Availability

All data generated or analyzed during this study are included in this published article.

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
