# Peer review of "Efficacy of Combined Initial Treatment of Methotrexate with Infliximab in Pediatric Crohn’s Disease: A Pilot Study"

_biomedicines, 2023, doi:10.3390/biomedicines11092575_

Round 1

Reviewer 1 Report (Previous Reviewer 1)

Revised manuscript has been rewrited according to reviewer7s comments.

In this revised manuscript authors have suggested the results will be of benefit to pediatric patients with Crohn's disease. Final conclusion has to be waited until much greater trial has done rightly. 

Author Response

Thank you for your comment. We have revised our text as advised. (see Page 1, line 39 and Page 12, line 417)

Reviewer 2 Report (Previous Reviewer 2)

The article presented by Yoon Zi Kim and collaborators, entitled “Efficacy of Combined Initial Treatment of Methotrexate with Infliximab in Pediatric Crohn´s Disease: a pilot study”, is a pilot sutdy that aimed to compare the efficacy and immunogenicity of methotrexate and azathioprine as concurrent therapies (Infliximab) for pediatric Crohn’s disease in South Korea population. The contribution of the manuscript to scientific literature is medium-low. The work is well written and is well understood, the figures and tables are correct. The objective is clear and the execution is correct. I only have minor comments

Minor revision:

1.       Line 56. ADAs can reduce the efficacy of anti-TNF-α agents by either neutralizing the bio-56 logic or facilitating the clearance of the drug. Reference

2.       Line 87. 15 mg/BSA. Acronym

3.       Line 302. Moreover, in our medical center, we 301 mainly applied a top-down strategy for treating pediatric CD. What does the top-down strategy consist of?

Author Response

1. Line 56. ADAs can reduce the efficacy of anti-TNF-α agents by either neutralizing the bio-56 logic or facilitating the clearance of the drug. Reference

Thank you for your comment. We have revised our text as advised. (see Page 2, line 57)

2. Line 87. 15 mg/BSA. Acronym

Thank you for your comment. We have revised our text as advised. (see Page 11, line 358)

3.Line 302. Moreover, in our medical center, we 301 mainly applied a top-down strategy for treating pediatric CD. What does the top-down strategy consist of?

Thank you for your comment. We have revised our text as advised. (see Page 17, line 301-303)

Reviewer 3 Report (Previous Reviewer 3)

Authors misinterpreted my criticism in point 7. The comment does not concern safety of MTX over AZA. Authors should simply explain why MTX was more effective than AZA in their cohort.

All other answers were fine.

Author Response

Thank you for your comment. In this pilot study, we aimed to provide information on the non-inferiority of MTX to AZA in terms of remission, and show that the immunogenicity of MTX is comparable to AZA, presenting MTX as a flexible option for immunomodulatory selection. I agree that whether MTX has advantages over thiopurine requires further studies in different research setting.

Reviewer 4 Report (Previous Reviewer 4)

As a reviewer of the previous version, I was very glad to see such massive improvement of the manuscript. I would like to emphasize that, in the Authors’ reply to my comments, the most frequent aspect was about particularities of common practice in “South Korea or in Asian population”, contrary to Western countries. E.g.

·       “most clinicians in real-life practice treating patients with Korean CD are more familiar with thiopurines and the usage of MTX is somewhat less common in many settings…,

·       “The typical treatment is starting with an AZA dose of 2.0-2.5mg/kg/day in Western countries, whereas dose-escalating strategy tend to be used in Asian countries to stratify AZA therapy starting with lower doses of 0.5-1.0mg/kg/day…”,

·       “in terms of maintenance therapy, there is cumulative evidence showing that therapeutic efficacy even at a low dose of AZA (1.0-1.5mg/kg/day) in treating IBD for Asian populations.”,

·       “It is true that guidelines for Crohn’s disease (CD) treatment mentioned that methotrexate can be used to maintain clinical remission as a first-choice immunomodulator. However, most clinicians in real-life practice, especially in South Korea, are still more familiar with thiopurines.”,

·       “In the Korea Health Insurance Review and Assessment Service, pediatric CD patients with PCDAI ≥30 must have a history of using at least two drugs among steroid, 5-ASA, or immunomodulator for at least 3 months in order for biologics to be covered by insurance.”,

·       “In Korea, PCDAI is still used in clinical practice because the National Insurance Policy requires PCDAI scores instead of wPCDAI scores.”,

·    “using MTX as the first –line concurrent medication in South Korea is still relatively low. Base on this local background, we aimed to provide information about non-inferiority of MTX to AZA in terms of remission, and even show that the immunogenicity of MTX is comparable to AZA, presenting MTX as a flexible option.”.

All these explanations were introduced by the Authors in this new version of the manuscript and everything is clear now and makes sense. Also, the Authors corrected their errors. The main limitation remains the very small number of patients, but we have to think that, if the paper is published, it will encourage physicians from South Korea and other Asian countries to use MTX. And, this is the whole purpose. MTX works and it should be used. I appreciated the insertion of the reference 14 (Ha et al), detailing about “Medication use and drug expenditure in inflammatory bowel disease: based on Korean National Health Insurance claims data”.

Some minor comments:

1. Results: The Authors wrote: “From January 2020 to January 2022, 34 patients were screened for the study. Of these, three patients who did not meet the inclusion criteria were excluded, and one patient was inevitably withdrawn from the study owing to insurance issues that prevented the continued infusion of IFX. In total, 29 patients were randomly assigned”… But, three plus one gives 4. What happened with the 5th patient? Since only 29 were included, not 30.

2. Discussion: The Authors wrote: “It has been observed that most side effects of MTX can be managed by reducing the dose, indicating that the occurrence of these events is independent of the dose [35]”. I think the Authors wanted to write “dependent” instead of “independent”. Please correct.

3. I suggest the recent reference by “Kappelman MD, et al. Comparative Effectiveness of Anti-TNF in Combination With Low-Dose Methotrexate vs Anti-TNF Monotherapy in Pediatric Crohn's Disease: A Pragmatic Randomized Trial. Gastroenterology. 2023 Jul;165(1):149-161.e7.” – to be inserted and commented on.

Generally, good quality.

Author Response

1. Results: The Authors wrote: “From January 2020 to January 2022, 34 patients were screened for the study. Of these, three patients who did not meet the inclusion criteria were excluded, and one patient was inevitably withdrawn from the study owing to insurance issues that prevented the continued infusion of IFX. In total, 29 patients were randomly assigned”… But, three plus one gives 4. What happened with the 5thpatient? Since only 29 were included, not 30.

Thank you for your comment. The number of patients excluded was incorrectly stated. We have revised our text as advised. (see Page 4, line 166)

2.Discussion: The Authors wrote: “It has been observed that most side effects of MTX can be managed by reducing the dose, indicating that the occurrence of these events is independent of the dose [35]”. I think the Authors wanted to write “dependent” instead of “independent”. Please correct.

Thank you for your comment. We have revised our text as advised. (see Page 11, line 358)

3. I suggest the recent reference by “Kappelman MD, et al. Comparative Effectiveness of Anti-TNF in Combination With Low-Dose Methotrexate vs Anti-TNF Monotherapy in Pediatric Crohn's Disease: A Pragmatic Randomized Trial. Gastroenterology. 2023 Jul;165(1):149-161.e7.” – to be inserted and commented on.

Thank you for your comment. We have revised our text as advised. (see Page 10, line 306 and Page 14, line 495-498 in Reference)

This manuscript is a resubmission of an earlier submission. The following is a list of the peer review reports and author responses from that submission.

Round 1

Reviewer 1 Report

I read the manuscript with much interest. A prospective randamised study design in this study is understandable but the number of cases is rather small;

1, Did authors calculated the minimum number need to provide non-inferiority of the two medication in this study?

2, Table 2 is missing please show the table.

3, AZA or MTX, which is prefered in pediatric CD patients in the first line? Please discuss much more deeply.

Author Response

1, Did authors calculated the minimum number need to provide non-inferiority of the two medication in this study?

Thank you for your comment. We agree that sample in this study is rather small. At the beginning, we calculated the minimum number need to provide non-inferiority of the two medication in this study. Unfortunately, there was an obstacle to enroll enough pediatric patients in this prospective study as parent’s concern about it.

2, Table 2 is missing please show the table.

As we describe about safe issue in main text (page 8, line 258-262), we delete Table 2.

3, AZA or MTX, which is preferred in pediatric CD patients in the first line? Please discuss much more deeply.

Thank you for your comment. In real-life practice, most clinicians treating pediatric patients with CD are more familiar with thiopurines and the use of MTX is somewhat less common in many settings because reports on combination treatment with MTX are relatively scarce. However, administration of AZA without genetic assessment and AZA monitoring systems carries the risk of malignancy or bone marrow suppression, leading to severe outcome such as sepsis or death. In this study, we aimed to provide information about non-inferiority of MTX to AZA in terms of remission and immunogenicity. Therefore, we suggest considering MTX as the first line concurrent treatment with IFX for pediatric patients with moderate-to-severe CD when thiopurine genetic assessment and AZA monitoring systems are not available. Whether MTX has advantages over thiopurine requires further studies in different research setting.

We have revised our text as advised. (see Page 1, line 35 in abstract and Page 11, line 383)

Reviewer 2 Report

The article presented by Yoon-Zi Kim and collaborates, entitled “Efficacy of Combined Initial Treatment of Methotrexate with Infliximab in Pediatric Crohn´s Disease”, is an original article that aimed to analyze the efficacy of an initial combination treatment of methotrexate and infliximab in comparison to azathioprine in pediatric Crohn´s Disease. The contribution of the manuscript to scientific literature is medium-low.

Major revision:

1.      Chart footers must contain the description. The graphics when working with patients is better to represent them with points to see the dispersion (Figure 1 and 2).

2.      Line 266. Nevertheless, the number of published studies on the ef-266 fect of MTX in achieving remission in moderate-to-severe CD is still limited compared to 267 that of AZA. Give examples.

Author Response

  1. Chart footers must contain the description. The graphics when working with patients is better to represent them with points to see the dispersion (Figure 1 and 2). 

Thank you for your comment. Figure 1 and 2 are showing the percentage of patients who reached remission. Rather than plotting each patients as a dot, the percentage was expressed as a bar graph.

  1. Line 266. Nevertheless, the number of published studies on the ef-266 fect of MTX in achieving remission in moderate-to-severe CD is still limited compared to 267 that of AZA. Give examples.

Thank you for your comment. To date, there have been no RCTs and almost all publications are on MTX use after thiopurine failure or intolerance. We have revised our text as advised. (see Page 9, line 301 with adding reference)

Reviewer 3 Report

In the present randomized study on pediatric patients with Crohn’s disease (CD), Kim et al showed that concurrent treatment with azathioprine (AZA) or methotrexate (MTX) is equally effective and has similar immunogenicity. Main comments:

1) Page 30: please use TL rather than through.

2) Page 2 line 59: check “thioprines”.

3) Dose of AZA was reported as 1 mg/kg in line 23 (abstract) and 0.5 mg/kg at line 84.

4) How was evaluated in the protocol the need of IFX dose escalation or reduction of administration interval?

5) Was steroids use allowed in any phase of the study?

6) In table 1, Authors should report disease behaviour and extension according to Montreal classification.

7) The Discussion about why MTX could be slightly more effective than AZA is poor and not convincing. A more in-depth analysis and discussion in necessary.

none

Author Response

1) Page 30: please use TL rather than through.

Thank you for your comment. However, I can’t find any “through” in this main text.

2) Page 2 line 59: check “thioprines”.

Thank you for your comment. I changed “thioprines” to “thiopurines”. We have revised our text as advised. (see Page 2, line 61)

3) Dose of AZA was reported as 1 mg/kg in line 23 (abstract) and 0.5 mg/kg at line 84.

Thank you for your comment. We have revised our text as advised. (see Page 1, line 23 (abstract)).

4) How was evaluated in the protocol the need of IFX dose escalation or reduction of administration interval?

Thank you for your comment. We didn’t allow IFX dose escalation or reduction of administration interval in this study. Because IFX dose escalation may affect remission. Dropout from the study was considered at the discretion of the researcher if IFX dose intensification was required by LOR during the study. We have revised our text as advised. (see Page 2, line 96-97)

5) Was steroids use allowed in any phase of the study?

Thank you for your comment. Steroid use was not permitted during this study because steroid may affect remission. However, subjects who were on steroid reduction at the time of enrollment were allowed to enroll in the study. We have revised our text as advised. (see Page 2, line 98-100)  

6) In table 1, Authors should report disease behavior and extension according to Montreal classification.

Thank you for your comment. We collected information by using Paris classification at diagnosis. We have revised our text as advised. (see Page 5, Table 1)

7) The Discussion about why MTX could be slightly more effective than AZA is poor and not convincing. A more in-depth analysis and discussion in necessary.

Thank you for your comment. In real-life practice, most clinicians treating pediatric patients with CD are more familiar with thiopurines and the use of MTX is somewhat less common in many settings because reports on combination treatment with MTX are relatively scarce. However, administration of AZA without genetic assessment and AZA monitoring systems carries the risk of malignancy or bone marrow suppression, leading to severe outcome such as sepsis or death. In this study, we aimed to provide information about non-inferiority of MTX to AZA in terms of remission and immunogenicity. Therefore, we suggest considering MTX as the first line concurrent treatment with IFX for pediatric patients with moderate-to-severe CD when thiopurine genetic assessment and AZA monitoring systems are not available. Whether MTX has advantages over thiopurine requires further studies in different research setting. We have revised our text as advised. (see Page 1, line 35 in abstract and Page 11, line 383)

Reviewer 4 Report

The authors performed a good study. MTX is always a very good option and they proved it (well, in a very small sample of patients - which, even if mentioned, still remains an issue). It has the advantage of being a prospective study. However, there are some remarks regarding their manuscript:

A. Abstract:

1. The authors wrote in Background: “The combination of antitumor necrosis factor-alpha (TNF-α) agents with immunomodulators, such as methotrexate (MTX) and azathioprine (AZA), is a common treatment for pediatric Crohn's disease (CD)”. If it is so common, why did they perform the study and write this manuscript? And why did they not cite all these references of scientific evidence in MTX case? Then, please rephrase this sentence in the “Background” and emphasize the strength of what you did.

2. Luminal remission – please define it better (parameters on imaging). Maybe use “transmural remission”.

3. It appears that AZA was used at a very low dose, while for MTX, the usual doses was used. Why? Please comment on this in “Discussion” or even in “Material and methods”.

4. About Conclusion: Rheenen et al: ESPGHAN/ECCO Guidelines (2020) stated that: “Methotrexate can be used to maintain clinical remission as a first-choice immunomodulator, or after thiopurine failure or intolerance. LoE: 3 | Agreement: 96%.” Therefore, why to be an alternative to AZA with IFX therapy? Also, it was stated that: ”Immunomodulators, including thiopurines and methotrexate, administered concomitantly with anti-TNF agents, reduce the likelihood of antidrug antibody [ADA] development”. Moreover, over the last years, in the US and Canada, AZA was used less and less, fear of HSTCL. Instead, MTX utilization increased greatly. Why this Conclusion then? I see this was not the initial text the authors wrote, but it was added to “MTX can be considered as an initial combination treatment with IFX“: “when thiopurine genetic assessment and AZA monitoring systems are not available”. Genetic testing for AZA is not universally available, the same for AZA monitoring metabolites levels. Thus, please remove the addition.

B. Introduction:

1. Line 42 – please delete “As a classic drug for the treatment of” and start with “In Crohn’s disease…,” as it does not make any sense. MTX is an old drug, but this sentence is about the combo. And please add new references.

2. Line 58 – please add “azathioprine (AZA)” to “6-MP”, as AZA is much more used in countries from Europe.

C. Material and methods:

a. Lines 91-91: “Mesalazine was administered at doses of 40–80 mg/kg/day concurrently with AZA and MTX in all patients”. Please explain. Why were they used? Not recommended, just in only mild colonic cases. Especially, when administering a biologic, continuation of 5-ASA did not show any benefit (please see recent publications). And, in any case it was a “placebo” dose (Table 1) – in Results.

b. Line 107: Why was disease activity assessed by PCDAI and not wPCDAI?

c. Statistical Analysis: Multivariate logistic regression? to adjust for potential confounders? I do not see it.

D. Results:

Table 1 is wrong. When showing classification by age, the total sum gives 30, not 29. If this gets wrong, how could we trust this study? Behavior also gives 30 patients!!! It is the AZA group – showing 15 patients, not 14! This mistake puts in question all the results!

E. Discussion:

1. This paragraph should start with the authors’ findings. Instead, there is a lot of theory, that could be used in Introduction.

2. As said, TPMT genotyping is not available in many countries. They still use MTX. If this manuscript is published, then thousands and thousands of children could not be treated with MTX, just because thiopurine genotyping and monitoring is not available. This is out of order.

F. References – incredibly old. Please update from the recent literature.

 Minor editing of English language required

Author Response

A. Abstract:

1. Thank you for your comment. We agree that the sentence, “The combination of antitumor necrosis factor-alpha (TNF-α) agents with immunomodulators, such as methotrexate (MTX) and azathioprine (AZA), is a common treatment for pediatric Crohn's disease (CD)”, could lead misunderstanding. Although MTX can be a first-line medication as an immunomodulators in Crohn’s disease, most clinicians in real-life practice treating patients with Korean CD are more familiar with thiopurines and the usage of MTX is somewhat less common in many settings (Immunomodulator usage in Korea report: AZA 92.7% vs MTX 1.8%). Therefore, we aimed to provide information about non-inferiority of MTX to AZA in terms of remission and immunogenicity. Finally, we suggest considering MTX as the first line concurrent treatment with IFX for pediatric CD when thiopurine genetic assessment and AZA monitoring systems are not available. Whether MTX has advantages over thiopurine requires further studies in different research setting. We have revised our text as advised. (see Page 1, line 21 to 22, page 2 line 61-64)

2. Thank you for your comment. We have revised our text as advised. (see Page 1, line 28, 32 in abstract,  Page 3; line 134, 138, Page 4 line 151, line 154, Page 6, line 198, Page 8, line 266, line 270, Page 10 line 295)

3. Thank you for your comment. The typical treatment is starting with an AZA dose of 2.0-2.5mg/kg/day in Western countries, whereas dose-escalating strategy tend to be used in Asian countries to stratify AZA therapy starting with lower doses of 0.5-1.0mg/kg/day, then subsequently carefully adjusts the dose to the target dose according to the thiopurine metabolite level as well as genetic testing results. The main reason for differences AZA dosing regimens in Asian countries is the high incidence of thiopurine-induced leukocytopenia in Asian patients, even at lower doses than those used in Western countries. Moreover, in terms of maintenance therapy, there is cumulative evidence showing that therapeutic efficacy even at a low dose of AZA (1.0-1.5mg/kg/day) in treating IBD for Asian populations. Similarly, in this study, even at the low dose of AZA maintained clinical remission with an adequate therapeutic level of thioprine metabolite. (from our study, mean 6-TGN level after 1 year of treatment was 245.2 ± 68.85 pmol/8â…¹108 RBC) We have revised our text as advised. (see Page 11, line 395 to 405)

4. Thank you for your comment. It is true that guidelines for Crohn’s disease (CD) treatment mentioned that methotrexate can be used to maintain clinical remission as a first-choice immunomodulator. However, most clinicians in real-life practice, especially in South Korea, are still more familiar with thiopurines. (Azathioprine use in CD was 92.7% vs methotrexate use in CD was 1.8%) Therefore, this study sought to demonstrate once again the efficacy of methotrexate as a first-line immunomodulator but also the immunogenicity such as protection in anti-drug antibody formation of methotrexate in pediatric Crohn’s disease. We have revised our text as advised. (see Page 1, line 21 to 22, page 2 line 61-64)

B. Introduction: 

1. Thank you for your comment. We have revised our text as advised. (see Page 1, line 44)

2. Thank you for your comment. We have revised our text as advised. (see Page 2, line 59)

C. Material and methods:

1. Thank you for your comment. In the Korea Health Insurance Review and Assessment Service, pediatric CD patients with PCDAI ≥30 must have a history of using at least two drugs among steroid, 5-ASA, or immunomodulator for at least 3 months in order for biologics to be covered by insurance. Because of this insurance practice, the patients in this study received 5-ASA and an immunomodulator when the diagnosis was confirmed. We have revised our text as advised. (see Page 2, line 97 to 100)

2. Thank you for your comment. In Korea, PCDAI is still used in clinical practice because the National Insurance Policy requires PCDAI scores instead of wPCDAI scores. Thus, we used PCDAI scores instead of wPCDAI scores.

3. Thank you for your comment. We additionally analyzed univariate Cox proportional hazard regression and revised our text as advised (see page 4, line 152-158; page 9, Table 2, Figure 5)

D. Results: 

1. Thank you for your comment. We have revised our text as advised. (see Page 5, Table 1)

E. Discussion:

1. Thank you for your comment. We have revised our text as advised. (see Page 10, line 292-297)

2. Thank you for your comment. I think the reviewer miswrite “MTX” instead of “AZA”, and there seems to be concern about limited use of AZA just because thiopurine genotyping and monitoring is not available. This study does not mean that we should not use thiopurines simply because the thiopurine genotyping and monitoring is not available. As I already mentioned above, using MTX as the first –line concurrent medication in South Korea is still relatively low. Base on this local background, we aimed to provide information about non-inferiority of MTX to AZA in terms of remission, and even show that the immunogenicity of MTX is comparable to AZA, presenting MTX as a flexible option. We have revised our references with the latest one. (see Page 1, line 38-39 in abstract and Page 12, line 414-416)

F. References:

1. Thank you for your comment. We have revised our references with the latest one. (see Page 12-14)

Round 2

Reviewer 1 Report

 Author have to think about sample size of this study. You have to refer to a literature below; in Gastroenterology 2023; 165: 149–161; Comparative Effectiveness of Anti-TNF in Combination With Low-Dose Methotrexate vs Anti-TNF Monotherapy in Pediatric Crohn’s Disease: A Pragmatic Randomized Trial Michael D. Kappelman, et al.

This article is dealing with the same issue as your paper. In the article they define their sample size: We estimated a necessary sample size of 353 participants and set a recruitment target of 425 participants to explore heterogeneity of treatment effects. As combination therapy with methotrexate may improve response, we performed a multicenter, randomized, double-blind, placebo-controlled pragmatic trial to compare tumor necrosis factor inhibitors with oral methotrexate to tumor necrosis factor inhibitor monotherapy.

 By the way this author's paper is ignored sample size necessity to be very small. 

Rethinking and rewriting is warranted.

no

Author Response

Thank you for your comment. We agree that sample in this study is rather small. Unfortunately, there was an obstacle to enroll enough pediatric patients in this prospective study as parent’s concern about it. Additionally, this study was a pilot study. Therefore, we have indicated in the title and the manuscript that this study was a pilot study. Further prospective study with larger sample size is required to reach more concrete conclusion. We have revised our text as advised. (see Page 11, line 393-394)

Reviewer 3 Report

Answers were satisfactory

Author Response

Thank you for your comment.